# Deciphering changes in the incidence of hemorrhagic stroke and cerebral venous sinus thrombosis during the coronavirus disease 2019 pandemic: A nationwide time-series correlation study

Soo Hyeon Cho[1], Dong Kyu Kim[2], Min Cheol Song[3], Euiho Lee[4,5], Seoncheol Park[6,7], Darda Chung[8]*, Jongmok Ha[9,10]*

1 Department of Nursing, Graduate School of Yonsei University, Seoul, Korea, 2 Department of Internal Medicine, Daegu Catholic University Medical Center, Daegu, Korea, 3 Health Policy Division, Public Health Center, Yangpyeong County Office, Yangpyeong, Korea, 4 Department of Integrative Medicine, Yonsei University College of Medicine, Seoul, Korea, 5 Department of Internal Medicine, Yongin Severance Hospital, Yongin, Korea, 6 Department of Mathematics, Hanyang University, Seoul, Korea, 7 Research Institute for Natural Sciences, Hanyang University, Seoul, Korea, 8 Department of Neurology, Korea University Anam Hospital, Seoul, Korea, 9 Department of Neurology, Samsung Medical Center, Sungkyunkwan University School of Medicine, Seoul, Korea, 10 Neuroscience Center, Samsung Medical Center, Seoul, Korea

☯ These authors contributed equally to this work.
* jongmok3245@gmail.com (JH); vogmag0107@gmail.com (DC)

**Editor:** Sonu Bhaskar, Global Health Neurology Lab / NSW Brain Clot Bank, NSW Health Pathology / Liverpool Hospital and South West Sydney Local Health District / Neurovascular Imaging Lab, Clinical Sciences Stream, Ingham Institute, AUSTRALIA

## Abstract

### Introduction

Hemorrhagic stroke and cerebral venous sinus thrombosis (CVST) are associated with severe acute respiratory syndrome coronavirus 2 (SARS-CoV-2) infection and vaccination. We aimed to investigate changes in the incidence of hemorrhagic stroke and CVST in South Korea before and during the coronavirus disease 2019 pandemic and the factors associated with these changes.

### Materials and methods

We conducted a nationwide time-series study using population-based databases between 2007 and 2022. The real-world and forecasted incidences of acute non-traumatic subarachnoid hemorrhage (SAH), intracerebral hemorrhage (ICH), and CVST during the pandemic period (2020–2022) were estimated and compared with the pre-pandemic period (2007–2019). The prevalence of conventional risk factors was measured using time-series data. Finally, a time-series correlation analysis was performed to examine the temporal association between conventional risk factors, SARS-CoV-2 infection, and SARS-CoV-2 vaccination.

**Data Availability Statement:** De-idenitified data used for this study is available at National Health Insurance Sharing Service (https://nhiss.nhis.or.kr) or HIRA Healthcare Big-data Hub (https://opendata. hira.or.kr) for approved studies.

**Funding:** The author(s) received no specific funding for this work.

**Competing interests:** The authors have declared that no competing interests exist.

## Results

The incidence of hemorrhagic stroke (SAH and ICH) was lower during the pandemic than during the pre-pandemic period. This observed decrease was associated with a reduction in the prevalence of conventional risk factors but not with SARS-CoV-2 infection or vaccination. The incidence of CVST was higher during the pandemic than during the pre-pandemic period, which may be temporally related to SARS-CoV-2 vaccination (Pearson correlation coefficient [r] = 0.349, $P$ = 0.031).

## Conclusion

We report reassuring evidence of hemorrhagic stroke associated with SARS-CoV-2 infection and vaccination. However, awareness of CVST may be required for future vaccine roll-outs and SARS-CoV-2 outbreaks.

## 1. Introduction

Various neurological complications have been associated with SARS-CoV-2 infection and vaccination, raising public health concerns [1, 2]. One of the controversial complications linked to these novel immune stimuli is hemorrhagic stroke and cerebral venous sinus thrombosis (CVST). These diseases are particularly relevant because they are major neurological conditions that can cause disabilities and societal burden [3].

Hemorrhagic stroke or intracranial hemorrhage can crudely be dichotomized into intracerebral hemorrhage (ICH) and subarachnoid hemorrhage (SAH), each of which has overlapping causes (e.g., hypertension, usage of antithrombotic medication) and distinct etiologic risk factors (e.g., arteriovenous malformation for the former and intracranial aneurysm for the latter) [4]. Moreover, CVST is a venous thrombotic disorder that can cause seizures and venous strokes in young adults and is often associated with the use of oral contraceptives or conditions that provoke serum hypercoagulability [5].

Data-driven, self-controlled case series studies have identified an association between hemorrhagic stroke and SARS-CoV-2 vaccines. A study in England has reported an increased risk of ICH, 1–7 and 15–21 days after the first dose of BNT162b2 mRNA-based vaccine, and an increased risk of SAH following 0–14 days of SARS-CoV-2 infection [6]. Another study in Wales has reported an increased risk of hemorrhagic events at 8–14 days after the first dose of the BNT162b2 vaccine and 0–28 days after SARS-CoV-2 infection [7]. Furthermore, a Scottish study has identified increased hemorrhagic risks within 7–27 days of the first ChAdOx1-S/ nCoV-19 vaccine dose and within 28 days of the first BNT162b2 vaccine dose [8].

Viral vector-based SARS-CoV-2 vaccination, especially ChAdOx1-S/nCoV-19 was associated with CVST, which can be generally explained through mechanisms of thrombosis with thrombocytopenia syndrome (TTS) [9–11]. A Scottish study has identified an increased risk of venous thromboembolism, 14–27 days and >28 days after the first dose of the ChAdOx1-S/nCoV-19 vaccine [8]. Another study in England has reported that the first dose of the ChAdOx1-S/nCoV-19 vaccine increased the risk of CVST by an incidence risk ratio of 4.01 within 8–14 days [12].

Although these studies provide sufficient evidence to suggest an association, none have validated these findings using a time-series study design, especially in the Korean population. Thus, this study aimed to investigate changes in the incidence of hemorrhagic stroke and CVST in South Korea before and during the coronavirus disease 2019 (COVID-19) pandemic. Furthermore, we explored the factors that could explain these changes.

## 2. Materials and methods

### 2.1. Study design and participants

This nationwide time-series correlation study used population-based data from the Health Insurance Review and Assessment Service (HIRA), National Health Insurance Service (NHIS), and Korea Disease Control and Prevention Agency (KDCA) databases over a period of 15.5 years (January 2007 to July 2022).

Patients with non-traumatic SAH, ICH, and CVST were screened using the International Classification of Diseases (ICD)-10 codes I60, I61, and I67.6, respectively, based on the HIRA research data (identifier: M20230314001). To ascertain the acuteness of onset and urgency of the condition, we screened patients who had visited the emergency room because of hemorrhagic stroke or CVST. This decision assumed that acute conditions were more feasible in explaining the temporal association with a risk factor than subacute or chronic cases, which are inclined to have a mixed etiologic nature. Moreover, we excluded patients who had been diagnosed with hemorrhagic stroke or CVST within 2 years to avoid recurrence and those who had been diagnosed with acute ischemic stroke (ICD-10 code: I63) within 30 days to avoid the inclusion of hemorrhagic transformation cases. The crude incidence rate (CIR) for each month was calculated using the monthly incidence of the target disease as the numerator and the monthly average population as the denominator [13].

### 2.2 Data collection: Patient-level data on risk factors

We collected patient-level data on diagnoses, sociodemographic characteristics, comorbidities, and medication history using HIRA claims data. These included primary diagnoses, subdiagnoses, medical services received, medical institutions visited, and healthcare costs, including co-payments.

The relevant comorbid risk factors per incident cases were collected for hemorrhagic stroke on hypertension (HTN; I10–I15, I67.4, O10–11, and O13–16), [14] atrial fibrillation (AF; I48), end-stage renal disease (ESRD; N18.5), liver cirrhosis (LC; K70.30, K70.31, K71.7, K74.60, K74.69, and K76.6), [15] Marfan syndrome (Q87.4), autosomal dominant polycystic kidney disease (ADPKD; Q61.2), Moyamoya disease (MMD; I67.5), intracranial aneurysm (IA; I67.1 and Q28.3), and arteriovenous malformation (AVM; Q28.2).

For CVST, a history of cancer (C00-C97), head trauma within 3 months (S02.0, S02.8, S02.9, S06, and S07), [16] sepsis within 3 months (A40.0-A41.9, R65.1, and R57.2), [17] and pregnancy-related outcomes within 3 months (O80–O84) were recorded. We also attained relevant medication history in hemorrhagic stroke and CVST as follows: antiplatelet agents (AP; aspirin, dipyridamole, triflusal, clopidogrel, ticagrelor, ticlopidine, prasugrel, cilostazol, sarpogrelate, and beraprost) and oral anticoagulants (OAC; warfarin, rivaroxaban, apixaban, edoxaban, dabigatran, enoxaparin, dalteparin, and nadroparin) for hemorrhagic stroke and hormonal agents (bazedoxifene acetate, estrogen, and tibolone) and oral contraceptives (ethinyl estradiol, drospirenone, and levonorgestrel) for CVST. The prevalence of risk factors and medication use in incident hemorrhagic stroke or CVST cases was calculated on a monthly basis.

### 2.3. Data collection: COVID-19 and vaccination data

Nationwide SARS-CoV-2 infection data from January 2020 to December 2022 were collected as monthly outbreak summaries from the KDCA database. Data on SARS-CoV-2 vaccination status from February 2021 to December 2022 were extracted from the NHIS database.

For vaccination data, the vaccination rate was specified for the following three different platforms to allow an in-depth analysis of the biological mechanism: mRNA-based (including

BNT162b2 and mRNA-1273), viral vector-based (including ChAdOx1 nCoV-19 and Ad26. COV2.S), and recombinant protein vaccines (NVX-CoV2373).

## 2.4. Statistical analysis

In this study, total sex- and age-specific crude CIRs were calculated per 100,000 persons for the pre-pandemic (2007–2019) and pandemic (2020–2022) periods. The standardized morbidity ratio (SMR) was calculated as the ratio between the observed value during the pandemic and the pre-pandemic reference value [13, 18]. The 95% confidence interval (CI) for SMR was estimated using Byar's approximation method [19].

Furthermore, the monthly prevalence of comorbid risk factors and medication history were estimated in patients with hemorrhagic stroke and CVST (cumulative risk factor or medication history count / incident hemorrhagic stroke or CVST cases). Prevalence data were compared between the pre-pandemic and pandemic periods. The traditional Wald method was used to examine the CI for the difference between proportions [20]. Finally, the weighted ranks of the risk factors were assessed considering the risk ratios collected from previous literature and used for crude comparisons between the pre-pandemic and pandemic periods [21–29].

For the time-series analysis, a polynomial regression model for hemorrhagic stroke and a simple regression model for CVST were selected to forecast the expected incidence and 95% CIs during the pandemic (2021–2022) period based on the baseline pre-pandemic (2007–2019) incidence. The best-fit forecast models were compared based on the Akaike information criterion (AIC) and Bayesian information criterion (BIC). Models with the lowest AIC and BIC values were selected for the study [30]. The odds ratio (OR) between the expected (forecasted) value and the observed value (number of hemorrhagic stroke and CVST cases) was used for comparison. The OR, excluding 1.0, was clinically significant at the 5% level.

The temporal association between hemorrhagic stroke/CVST and prevalent comorbid risk factors, medication history, SARS-CoV-2 infection, and SARS-CoV-2 vaccination was determined using Pearson's correlation analysis of data collected during the pre-pandemic (2007–2019) and pandemic (2020–2022) periods. The degree of correlation followed a conventional paradigm (weak: $0.10 \leq r < 0.30$, moderate: $0.30 \leq r < 0.50$, and strong: $r \geq 0.50$) [31].

## 2.5. Standard protocol approval, registration, and patient consent

The Institutional Review Board at Samsung Medical Center granted an exemption for review of this study because it involved the analysis of de-identified data already obtained through the epidemiological investigation, presented a minimal risk to the participants, and met the needs of current public health interests (identifier: SMC 2023-03-056). The need for consent was waived owing to the retrospective nature of this study, which used de-identified population data. All the experiments were conducted in accordance with the Strengthening the Reporting of Observational Studies in Epidemiology Reporting Guidelines.

## 3. Results

### 3.1. Summary of collected data for analysis

A total of 2,428,946 cases were collected for I60, and 5,906,398 cases were collected for I61. Among these, emergency cases were 156,783 cases for I60, 309,116 for I61, and 1,022 for I67.6. Altogether, 6,847 patients with stage I60 and 23,866 patients with stage I61 were excluded because they were diagnosed with stage I63 within 30 days. Moreover, 33,961, 34,323, and 75 cases of I60, I61, and I67.6, respectively, were excluded from the study owing to 2-year

washout. Finally, 115,975 cases of I60, 250,927 cases of I61, and 947 cases of I67.6 were used for statistical analysis (Fig 1).

### 3.2. Incidence of hemorrhagic stroke (SAH and ICH) and CVST in pre-pandemic (2007–2019) versus pandemic (2020–2022) period

For SAH, the pre-pandemic CIR was 0.61 per 100,000 persons, whereas the pandemic CIR was slightly lower at 0.55 per 100,000 persons (SMR: 0.90; 95% CI, 0.89–0.91). The same downward trend in the incidence was observed in both sexes during the pandemic. When stratified by age, significantly lower CIR was observed for individuals aged ≥30 years during the pandemic period.

For ICH, the pre-pandemic CIR was 1.32 per 100,000 persons, whereas during the pandemic period, CIR decreased to 1.21 per 100,000 persons (SMR: 0.91; 95% CI, 0.91–0.92). A similar trend was observed for the sex-specific CIR in patients with ICH. Lower CIRs were observed in all age groups during the pandemic.

For CVST, the pre-pandemic CIR was 0.004 per 100,000 persons, whereas during the pandemic period, the CIR increased to 0.009 per 100,000 persons (SMR: 2.312; 95% CI, 2.018–2.650). The same trend was observed for the sex-specific CIR. When broken down into age groups, a higher CIR was observed in the 10–79-year age group (Table 1).

### 3.3. Real-world incidence of hemorrhagic stroke and CVST during the pandemic (2007–2019) period versus forecasted estimate

When spread by months, the observed incidence of SAH was significantly lower than the expected incidence in March, October, and December 2020; August and October–December 2021; January–April and July 2022 (rate ratios: 0.80–0.88). The observed incidence of ICH was within the expected incidence in all pandemic months except for March 2020 and March 2022 (rate ratios: 0.88 and 0.84, respectively). For CVST, the observed incidence was significantly higher than the forecasted incidence for May–July and October 2021, and June 2022 (rate ratios: 1.66–2.16) (Fig 2, S1 File).

### 3.4. Comparing the prevalence of conventional risk factors of hemorrhagic stroke and CVST during pre-pandemic (2007–2019) and pandemic (2020–2022) periods

During the pandemic period, a significantly higher prevalence of AP or OAC therapy, HTN, ESRD, AF, LC, ADPKD, MMD, and IA was observed among incident patients with SAH than during the pre-pandemic period.

Moreover, a notably higher prevalence of AP or OAC therapy, HTN, ESRD, AF, LC, ADPKD, MMD, and IA was observed in patients with incident ICH during the pandemic period than during the pre-pandemic period. Conversely, a notably lower prevalence of AVM was observed in patients with incident ICH during the pandemic compared with the pre-pandemic prevalence.

Finally, there was no significant difference between the prevalence of contraceptive and hormonal therapy, sepsis within 3 months, and cancer within one year in incident CVST cases (Fig 3, S1 File).

### 3.5. Time-series correlation analysis: Hemorrhagic stroke/CVST incidence, comorbid risk factor prevalence, SARS-CoV-2, SARS-CoV-2 vaccination

During the pre-pandemic period, prevalence of ESRD showed a mild negative temporal association (r = -0.151, P = 0.039) and intracranial aneurysm a mild positive temporal association

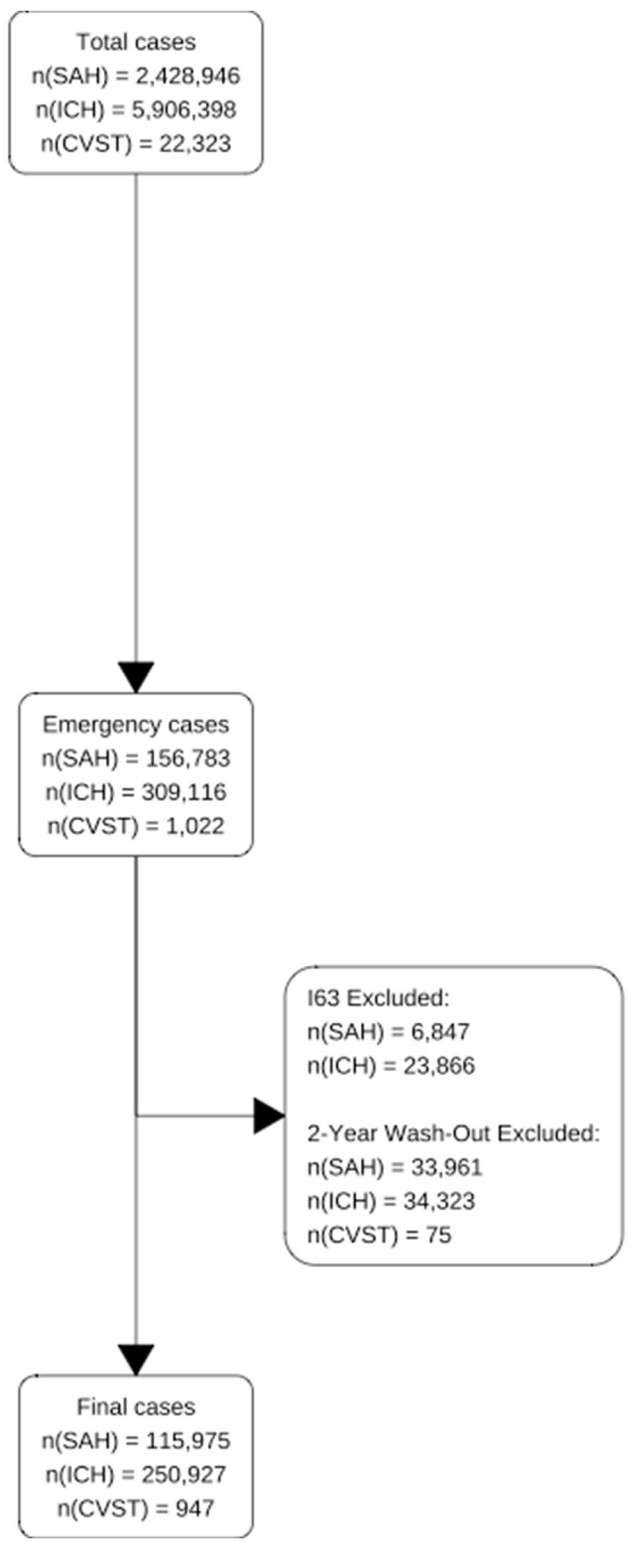

**Fig 1. Flow diagram of study participants.**

**Table 1. Sex-specific and age-specific crude cumulative incidence and rate ratio of hemorrhagic stroke and CVST during the pre-pandemic (2007–2019) and pandemic (2020–2022) periods.** Statistically significant results were bold typed.

| | | 2007–2019 (pre-pandemic) | | 2020–2022 (pandemic) | | pre-pandemic vs pandemic |
|---|---|---|---|---|---|---|
| | | N | CIR | N | CIR | SMR (95% CI) |
| SAH | Total | 97,958 | 0.61 | 18,017 | 0.55 | **0.9 (0.89–0.91)** |
| | Sex | | | | | |
| | Male | 39,484 | 0.50 | 6,680 | 0.41 | **0.83 (0.81–0.85)** |
| | Female | 58,474 | 0.73 | 11,337 | 0.68 | **0.94 (0.92–0.96)** |
| | Age | | | | | |
| | ≤9 | 258 | 0.00 | 40 | 0.00 | 0.78 (0.56–1.08) |
| | 10~19 | 623 | 0.07 | 92 | 0.06 | 0.87 (0.70–1.08) |
| | 20~29 | 1,770 | 0.17 | 265 | 0.15 | 0.89 (0.78–1.01) |
| | 30~39 | 6,396 | 0.56 | 784 | 0.38 | **0.67 (0.62–0.72)** |
| | 40~49 | 19,256 | 1.46 | 2,622 | 1.12 | **0.77 (0.74–0.80)** |
| | 50~59 | 25,644 | 1.96 | 4,410 | 1.64 | **0.84 (0.81–0.87)** |
| | 60~69 | 19,135 | 2.04 | 4,284 | 1.71 | **0.84 (0.81–0.86)** |
| | 70~79 | 16,164 | 2.79 | 3,007 | 1.98 | **0.71 (0.68–0.74)** |
| | ≥80 | 8,712 | 1.39 | 2,513 | 1.25 | **0.91 (0.87–0.95)** |
| ICH | Total | 211,386 | 1.32 | 39,541 | 1.21 | **0.91(0.91–0.92)** |
| | Sex | | | | | |
| | Male | 114,902 | 1.45 | 21,403 | 1.33 | **0.92(0.90–0.93)** |
| | Female | 96,484 | 1.20 | 18,138 | 1.10 | **0.91(0.90–0.93)** |
| | Age | | | | | |
| | ≤9 | 1,396 | 0.02 | 111 | 0.01 | **0.40(0.33–0.48)** |
| | 10~19 | 1,946 | 0.23 | 225 | 0.16 | **0.68(0.59–0.78)** |
| | 20~29 | 3,386 | 0.33 | 466 | 0.27 | **0.81(0.74–0.90)** |
| | 30~39 | 8,950 | 0.79 | 1,291 | 0.62 | **0.79(0.75–0.84)** |
| | 40~49 | 27,076 | 2.05 | 3,709 | 1.59 | **0.77(0.75–0.80)** |
| | 50~59 | 46,006 | 3.51 | 7,424 | 2.77 | **0.79(0.77–0.81)** |
| | 60~69 | 43,208 | 4.62 | 8,803 | 3.51 | **0.76(0.74–0.78)** |
| | 70~79 | 47,448 | 8.20 | 8,488 | 5.59 | **0.68(0.67–0.70)** |
| | ≥80 | 31,970 | 5.09 | 9,024 | 4.51 | **0.89(0.87–0.91)** |
| CVST | Total | 643 | 0.004 | 304 | 0.009 | **2.31(2.02–2.65)** |
| | Sex | | | | | |
| | Male | 340 | 0.004 | 169 | 0.010 | **2.45(2.03–2.94)** |
| | Female | 303 | 0.004 | 135 | 0.008 | **2.17(1.77–2.65)** |
| | Age | | | | | |
| | ≤9 | 12 | 0.000 | 2 | 0.000 | 0.83(0.22–3.24) |
| | 10~19 | 23 | 0.003 | 12 | 0.008 | **3.07(1.54–6.09)** |
| | 20~29 | 93 | 0.009 | 54 | 0.031 | **3.44(2.46–4.80)** |
| | 30~39 | 124 | 0.011 | 44 | 0.021 | **1.94(1.38–2.74)** |
| | 40~49 | 125 | 0.009 | 51 | 0.022 | **2.31(1.67–3.19)** |
| | 50~59 | 111 | 0.008 | 69 | 0.026 | **3.03(2.25–4.09)** |
| | 60~69 | 82 | 0.009 | 41 | 0.016 | **1.87(1.28–2.71)** |
| | 70~79 | 47 | 0.008 | 21 | 0.014 | **1.70(1.02–2.84)** |
| | ≥80 | 26 | 0.004 | 10 | 0.005 | 1.21(0.59–2.47) |

Crude CIR was calculated per 100 000 population.

‡The standardized morbidity ratio was calculated as (2020–2022 mean CIR)/(2007–2019 mean CIR).

Abbreviations: N, absolute number of patients; CIR, cumulative incidence rate; SMR, standardized morbidity ratio; SAH, subarachnoid hemorrhage; ICH, intracerebral hemorrhage; CVST, cerebral venous sinus thrombosis.

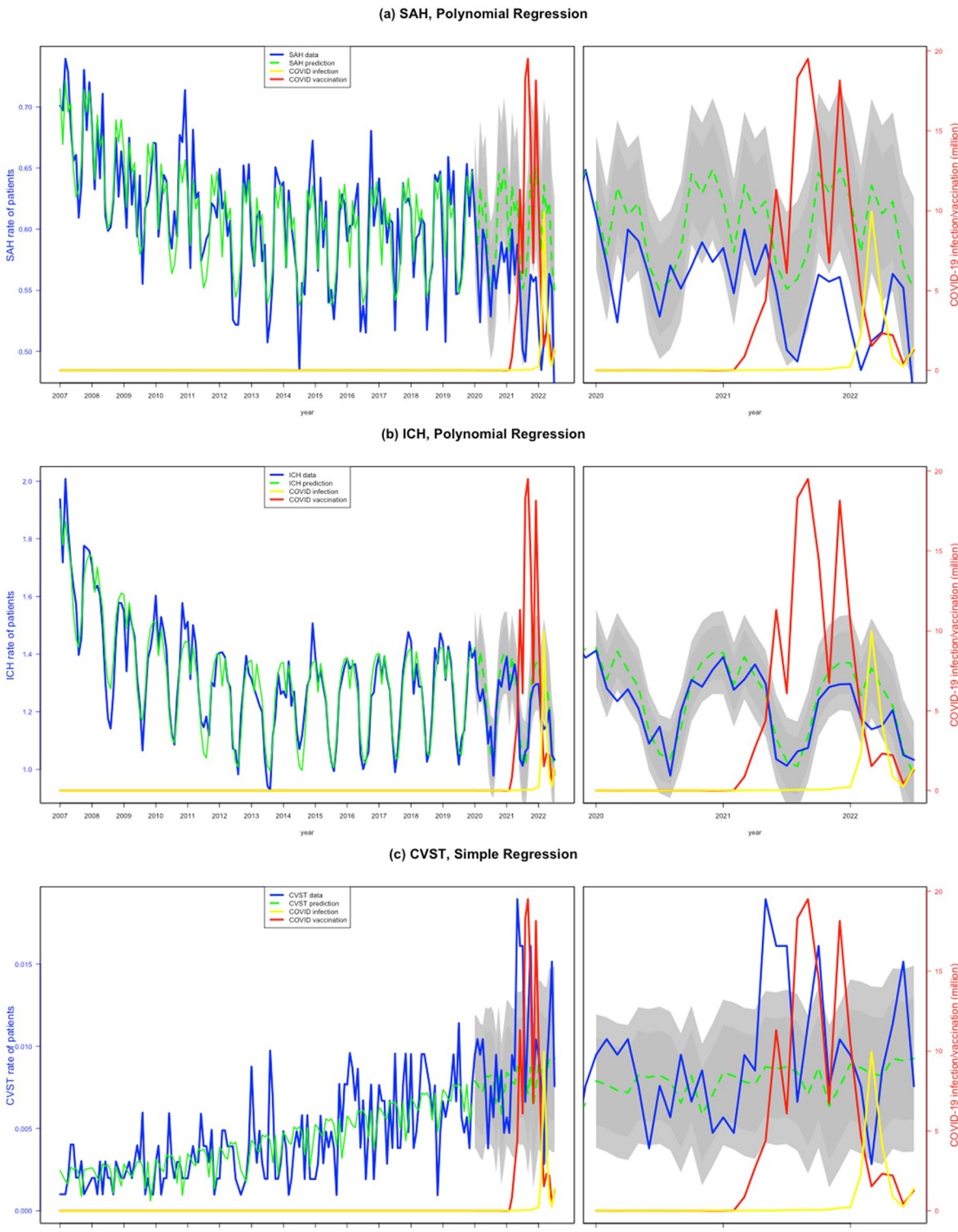

**Fig 2. Real-world incidence of hemorrhagic stroke and cerebral venous sinus thrombosis from 2020 to 2022 versus estimated incidence projections based on 2007–2019 incidence data.** The blue lines denote the observed incidence, green line the expected incidence, darker and lighter grey shades 80% and 95% CI of the predicted incidence, respectively. (a) subarachnoid hemorrhage (SAH). (b) intracerebral hemorrhage (ICH). (c) cerebral venous sinus thrombosis (CVST).

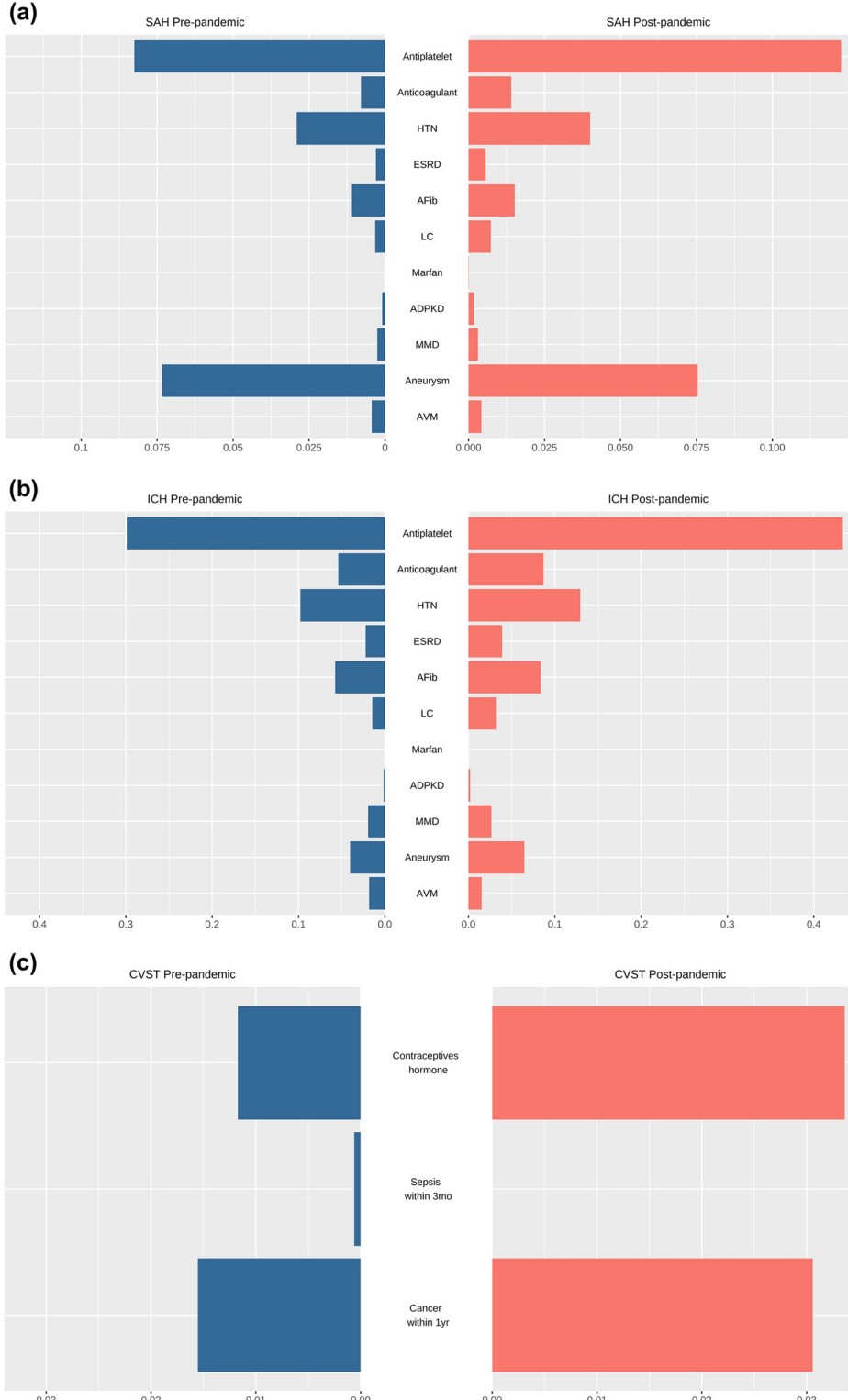

**Fig 3. Visualization of known risk factors for the prevalence of hemorrhagic stroke and cerebral venous sinus thrombosis (CVST) from the pre-pandemic (2007–2019) versus post-pandemic (2020–2022) period through a pyramid chart.** Blue bars in the left indicate pre-pandemic prevalence of incident hemorrhagic stroke or CVST cases. Red bars in the right indicate pandemic prevalence of incident hemorrhagic stroke or CVST cases. The x-axis denotes prevalence metrics per 100,000 persons. (a) subarachnoid hemorrhage (SAH). (b) intracerebral hemorrhage (ICH). (c)

cerebral venous sinus thrombosis (CVST). Abbreviations: HTN, hypertension; ESRD, end-stage renal disease; Afib, atrial fibrillation; LC, liver cirrhosis; ADPKD, autosomal dominant polycystic kidney disease; MMD, Moyamoya disease; AVM, arteriovenous malformation.

(r = 0.222, $P$ = 0.005) with SAH incidence. However, during the pandemic period, AP therapy and intracranial aneurysms showed a strong positive temporal association (r = 0.528 and r = 0.650; $P$ = 0.007 and $P$ < 0.001, respectively) with the incidence of SAH. Additionally, a moderately positive temporal association was found between SAH incidence and the prevalence of HTN, AF, and MMD (r = 0.422, 0.368, and 0.357; $P$ = 0.048, 0.021, and 0.016, respectively) during the pandemic period.

During the pre-pandemic period, the prevalence of OAC therapy, ESRD, AF, LC, and MMD showed a mild-to-moderate negative association with ICH. However, the prevalence of AVM was moderately positively associated with the incidence of ICH (r = 0.305, $P$ < 0.001). During the pandemic period, the prevalence of AP therapy and HTN demonstrated strong positive temporal associations with the incidence of ICH (r = 0.890 and 0.835, respectively; $P$ < 0.001 for both).

Finally, during the pre-pandemic period, the prevalence of contraceptive and hormonal therapies demonstrated a mildly positive temporal association with CVST incidence (r = 0.281, $P$ = 0.002). During the pandemic period, this association was no longer significant, and CVST incidence demonstrated a moderately positive temporal association with COVID-19 vaccination (r = 0.349, $P$ = 0.031) (Table 2).

## 4. Discussion

In this study we analyzed the incidences of new-onset hemorrhagic stroke and CVST before and after the COVID-19 pandemic. The observed incidence of hemorrhagic stroke was lower than expected during the pandemic period in certain months, and this trend was associated with a decrease in the diagnosis of conventional risk factors. The observed incidence of CVST was higher than expected during the pandemic period, and this trend may have been temporally associated with SARS-CoV-2 vaccination.

The incidence of SAH and ICH was lower during the pandemic period than during the pre-pandemic period. Owing to the affinity of SARS-CoV-2 to angiotensin-converting enzyme 2 prevalent in cerebrovascular endothelium, [32] the association between SARS-CoV-2 and acute ischemic stroke has been implicated by several studies [33, 34]. However, the association between SARS-CoV-2 and hemorrhagic stroke has been unclear [35, 36]. Our result is in line with large population studies that have reported lower volume of hospitalization or incidence of SAH or ICH in patients during the pandemic [37, 38]. This finding may be partly associated with hospital avoidance and saturation of emergency medical services during the height of the pandemic; specifically, during the peak delta variant outbreak in September 2021 and Omicron variant outbreak in January 2022 in South Korea. In part, further investigations into excess sudden unexpected deaths or deaths of unknown cause during the pandemic period may support our findings regarding the aftermath of lower detection of subarachnoid hemorrhage (SAH), which could have been rapidly fatal if left untreated. Additionally, this finding can be attributed to a decrease in the number of minor stroke cases, which do not exceed the health-care-seeking behavior threshold [39, 40].

Despite large-scale reports on the potential causative association between hemorrhagic stroke and mRNA-based SARS-CoV-2 vaccines, our study did not reveal any temporal associations between the two. Furthermore, despite the increased mean prevalence of conventional

**Table 2. Summary of time-series correlation analyses between hemorrhagic stroke & CVST and known risk factors during the pre-pandemic (2007–2019) and pandemic (2020–2022) periods.** Statistically significant results were bold typed.

| | | 2007–2019 (pre-pandemic) | | 2020–2022 (pandemic) | |
|---|---|---|---|---|---|
| | | r | p | r | p |
| SAH | Antiplatelet therapy | -0.055 | 0.500 | **0.528** | **0.007** |
| | Anticoagulant therapy | 0.051 | 0.470 | 0.177 | 0.345 |
| | Hypertension | -0.056 | 0.490 | **0.422** | **0.048** |
| | End-stage renal disease | **-0.151** | **0.039** | -0.155 | 0.432 |
| | Atrial fibrillation | 0.022 | 0.760 | **0.368** | **0.021** |
| | Liver cirrhosis | -0.127 | 0.094 | 0.115 | 0.543 |
| | Marfan syndrome | 0.061 | 0.432 | 0.064 | 0.394 |
| | ADPKD | 0.006 | 0.927 | -0.024 | 0.900 |
| | Moyamoya disease | 0.071 | 0.408 | **0.357** | **0.016** |
| | Intracranial aneurysm | **0.222** | **0.005** | **0.650** | **<0.001** |
| | Arteriovenous malformation | 0.028 | 0.711 | 0.131 | 0.523 |
| | SARS-CoV-2 infection | - | - | -0.393 | 0.058 |
| | SARS-CoV-2 vaccination | - | - | -0.279 | 0.093 |
| | mRNA-based | - | - | -0.264 | 0.056 |
| | Viral vector-based | - | - | -0.134 | 0.478 |
| | Recombinant protein | - | - | -0.359 | 0.095 |
| ICH | Antiplatelet therapy | -0.192 | 0.054 | **0.890** | **<0.001** |
| | Anticoagulant therapy | **-0.240** | **0.009** | 0.154 | 0.433 |
| | Hypertension | -0.132 | 0.179 | **0.835** | **<0.001** |
| | End-stage renal disease | **-0.335** | **<0.001** | 0.176 | 0.346 |
| | Atrial fibrillation | **-0.246** | **0.011** | 0.438 | 0.052 |
| | Liver cirrhosis | **-0.190** | **0.017** | 0.389 | 0.089 |
| | Marfan syndrome | 0.015 | 0.867 | 0.252 | 0.170 |
| | ADPKD | -0.102 | 0.119 | 0.137 | 0.567 |
| | Moyamoya disease | **-0.177** | **0.028** | 0.072 | 0.656 |
| | Intracranial aneurysm | -0.030 | 0.703 | 0.233 | 0.146 |
| | Arteriovenous malformation | **0.305** | **<0.001** | 0.294 | 0.099 |
| | SARS-CoV-2 infection | - | - | -0.179 | 0.111 |
| | SARS-CoV-2 vaccination | - | - | -0.185 | 0.321 |
| | mRNA-based | - | - | -0.094 | 0.586 |
| | Viral vector-based | - | - | -0.315 | 0.191 |
| | Recombinant protein | - | - | -0.247 | 0.135 |
| CVST | Contraceptives and hormones ≤ 90d | 0.281 | **0.002** | 0.047 | 0.699 |
| | Sepsis within ≤ 90d | 0.026 | 0.329 | - | - |
| | Trauma within ≤ 90d | - | - | - | - |
| | Cancer within ≤ 1y | 0.202 | 0.070 | 0.262 | 0.260 |
| | SARS-CoV-2 infection | - | - | -0.311 | 0.256 |
| | SARS-CoV-2 vaccination | - | - | 0.349 | **0.031** |
| | mRNA-based | - | - | 0.289 | 0.061 |
| | Viral vector-based | - | - | 0.297 | 0.247 |
| | Recombinant protein | - | - | -0.319 | 0.118 |

Note: Statistical significance was set at $P < 0.05$.

Abbreviations: r, Pearson's correlation coefficient; ADPKD, Autosomal Dominant Polycystic Kidney Disease; SAH, subarachnoid hemorrhage; ICH, intracerebral hemorrhage; CVST, cerebral venous sinus thrombosis

risk factors in incident hemorrhagic stroke cases during the pandemic (i.e., higher dependability on these risk factors), the prevalence decreased during the pandemic, resulting in a positive temporal association with the declining incidence of hemorrhagic stroke. By contrast, no significant temporal association was observed between hemorrhagic stroke incidence and SARS-CoV-2 infection or vaccination, indicating that the decreased incidence of hemorrhagic stroke during the pandemic was more attributable to the downward trend in the prevalence of conventional risk factors than to SARS-CoV-2 infection or vaccination. Alternatively, the decreased incidence of hemorrhagic stroke during the pandemic may have contributed to the lower discovery rate of important conventional risk factors in incident cases. Nevertheless, the decreased prevalence of certain conventional risk factors (i.e., HTN, AP therapy, and AF) is not confined to incident hemorrhagic stroke cases and represents a larger nationwide trend, including the general population, supporting the initial hypothesis [41–43].

When hemorrhagic stroke was categorized into SAH and ICH, the association between the prevalence of intracranial aneurysms and the decrease in SAH incidence increased from mild to strong during the pandemic, implying a stronger link. Several interpretations are possible. First, a notable portion of patients who experience SAH are often diagnosed with the culprit aneurysm only at the time of stroke diagnosis. Hence, this finding may indicate a decrease in the number of undiagnosed aneurysmal SAH during the pandemic. Alternatively, this finding may be a result of decreased diagnosis of unruptured incidental intracranial aneurysms owing to lower access to healthcare during the pandemic, leading to low SAH awareness and suboptimal health-seeking behavior, [44] especially in conjunction with the prevalence trends of AP therapy, HTN, AF, and MMD, which all demonstrated a downward trend and a significant positive temporal association with SAH incidence. When analyzed by weighted ranks of attribution, no significant change in ranks in the conventional risk factor domain was recognized before and after the pandemic, implying a lack of intragroup alterations and a stable contribution of different mechanisms in patients with SAH. This indicates a quantitative game, not a qualitative game.

For CVST, we identified an association between vector-based vaccines and increased incidence, in agreement with previous studies that utilize different research designs [12]. Viral vector-based vaccines have reported to be associated with TTS-related CVST [9]. However, in our study, when divided into vaccine platforms, all three mechanisms (mRNA-based, viral vector-based, and protein recombinant) demonstrated a positive temporal association with CVST incidence, although no significant association was observed. Case reports of CVST following mRNA-based vaccines have been recognized, [45, 46] implying the possibility of diverse biological mechanisms that may contribute to the development of CVST in vaccine recipients. Nevertheless, because of the limitations of our study, we believe that these findings should be interpreted with caution.

Our study has some limitations. First, observational time-series studies can only reveal possible temporal associations and not causation. Secondly, population-based data were not equally implicated for each individual, thus limiting the scope of interpretation. Third, we only included patients who visited the emergency medicine service, thereby limiting the number of patients evaluated in our study. However, this process likely ascertained the acute-onset patients who may have a higher chance of being linked to the risk factors. Furthermore, our list of conventional risk factors was not exhaustive, and other factors may have influenced the change in hemorrhagic stroke (e.g. smoking, diabetes, cerebral amyloid angiopathy, etc.) or CVST (e.g. vasculitis, prothrombotic hematological disorders, etc.) incidence. Regarding CVST, the number of reported events were relatively small compared to the overall variable count. Additionally, we have not yet examined post-vaccination period statistics, which could offer further insights into the contribution of vaccines to CVST. In future studies, we may

address these limitations by expanding the scope of venous thrombotic disorders beyond CVST, relaxing the inclusion criteria to encompass urgent outpatient clinic visits, and incorporating data from the post-pandemic period.

In conclusion, we offer insights into deciphering the changes in hemorrhagic stroke and CVST incidence over the years amidst the pandemic and its association with various conventional risk factors, SARS-CoV-2 infection, and SARS-CoV-2 vaccines. We believe that our study facilitates the public's understanding of a prevalent, notable cerebrovascular disease and provides reassuring evidence on the association between hemorrhagic stroke (SAH and ICH) and SARS-CoV-2 vaccines or infection, whereas reinforcing the already known association between SARS-CoV-2 vaccines and CVST for future vaccine rollout.

## Supporting information

**S1 File.**
(DOCX)

## Acknowledgments

The authors would like to acknowledge every frontline healthcare professional in the field who has strived to achieve the best patient outcomes during the tumultuous phase of the COVID-19 pandemic. We also thank Hyoseon Jeong for assistance with the pharmacological data search, which was crucial to our study.

## Author Contributions

**Conceptualization:** Dong Kyu Kim, Min Cheol Song, Euiho Lee, Darda Chung, Jongmok Ha.

**Data curation:** Soo Hyeon Cho, Seoncheol Park.

**Formal analysis:** Soo Hyeon Cho, Seoncheol Park.

**Investigation:** Dong Kyu Kim, Min Cheol Song, Euiho Lee, Darda Chung, Jongmok Ha.

**Methodology:** Seoncheol Park, Darda Chung, Jongmok Ha.

**Resources:** Seoncheol Park.

**Software:** Seoncheol Park.

**Supervision:** Darda Chung, Jongmok Ha.

**Validation:** Min Cheol Song, Euiho Lee.

**Writing – original draft:** Soo Hyeon Cho, Dong Kyu Kim.

**Writing – review & editing:** Seoncheol Park, Darda Chung, Jongmok Ha.

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
