## [Decision Letter · Decision Letter 0]

27 May 2024

PONE-D-24-10134Deciphering changes in the incidence of hemorrhagic stroke and cerebral venous sinus thrombosis during the coronavirus disease 2019 pandemic: a nationwide time-series correlation studyPLOS ONE

Dear Dr. Ha,

Thank you for submitting your manuscript to PLOS ONE. After careful consideration, we feel that it has merit but does not fully meet PLOS ONE’s publication criteria as it currently stands. Therefore, we invite you to submit a revised version of the manuscript that addresses the points raised during the review process.

We have received reports from the reviewers and have carefully considered their feedback. Based on this review, we would like to invite you to revise your manuscript. Please provide a point-by-point rebuttal to the reviewers' suggestions and comments.

We look forward to receiving the revised version of your manuscript.

We look forward to receiving your revised manuscript.

Kind regards,

Sonu Bhaskar, MD PhD FANA

Academic Editor

PLOS ONE

Journal Requirements:

**Additional Editor Comments:**

Thank you for submitting your work to PLOS One. We have received reports from the reviewers and have carefully considered their feedback. Based on this review, we would like to invite you to revise your manuscript. Please provide a point-by-point rebuttal to the reviewers' suggestions and comments.

We look forward to receiving the revised version of your manuscript.

Reviewers' comments:

Reviewer's Responses to Questions

**Comments to the Author**

1. Is the manuscript technically sound, and do the data support the conclusions?

Reviewer #1: Yes

Reviewer #2: No

2. Has the statistical analysis been performed appropriately and rigorously? 

Reviewer #1: Yes

Reviewer #2: Yes

3. Have the authors made all data underlying the findings in their manuscript fully available?

Reviewer #1: Yes

Reviewer #2: Yes

4. Is the manuscript presented in an intelligible fashion and written in standard English?

Reviewer #1: Yes

Reviewer #2: Yes

5. Review Comments to the Author

Reviewer #1: Thank you for the article.

The results of this study are not much different than what we expected from literature and clinical experience.

There are of course confounding factors that plague all studies related to COVID studies including this one:

-The vaccines became available very quickly from a historical perspective. This makes interpretation of temporal associations between when the pandemic and vaccination difficult. This is especially important if there is delay in reporting.

-The study relies on reported risk factors. This seems to be dependent on bias of health care professionals. For instance, oral contraceptives as a risk factor for CVST seem to be exaggerated. Not surprising then, that women are more often imaged for CVST than men, and less often that scans show CVST.

-After becoming apparent that some vaccines are associated with ICH, they were withdrawn at least for younger populations in some European countries. How was that in Korea?

-Definition of pandemic period can be discussed. By spring-summer of 2021 the number of severe cases had dropped drastically and for practical purposes the pandemic was mostly a common cold being diagnosed in asymptomatic individuals.

The authors claim that the decrease in incidence of SAH might be related to less access to health care. While there is evidence for some conditions such as appendicitis in March-May 2020 in Europe, by June 2020 it was business as usual for us in Europe. In the case of appendicitis as an example there was increased incidence of complications in the ensuing months with 20-40% presenting with abscess. How would that be for SAH if there is a delay?

CVST caused by vaccine is much more serious than a CVST caused by natural infections from an ethical point of view. This is also important since the risk for thrombosis with COVID infection seems to have been exaggerated based on what we know.

Reviewer #2: The authors present a retrospective cohort study evaluating the rates of SAH, ICH, and CVST in South Korea from 2007 - 2022. Rates of cancer, sepsis, hormone therapy, anticoagulation, aneurysm, HTN, renal disease, liver disease, etc. were also reported. The manuscript concludes that SAH and ICH rates declined during the pandemic, while CVST increased. The authors posit the conjecture that this may be related SARS-COV vaccination.

1) Please, replace the ICD-10 codes in your figures / tables with the names of the diseases.

2) I would recommend expanding your discussion around CVST. Particularly, it would be helpful to further discuss the limitations of your data (small N for CVST events versus model variable count, risk factors for CVST missing from dataset, lack of post vaccination push data...) and what additional studies you feel would be needed to better address this question.

6. PLOS authors have the option to publish the peer review history of their article (what does this mean?). If published, this will include your full peer review and any attached files.

Reviewer #1: **Yes: **Michael R. Torkzad

Reviewer #2: No

---

## [Author Response · Author response to Decision Letter 0]

16 Jun 2024

June 2nd, 2024

Dear Dr. Bhaskar

Re: Manuscript ID PONE-D-24-10134

Title: Deciphering changes in the incidence of hemorrhagic stroke and cerebral venous sinus thrombosis during the coronavirus disease 2019 pandemic: a nationwide time-series correlation study

On behalf of the authors, We wish to express our gratitude toward the reviewers' meticulous review of our manuscript. We have made revisions according to the reviewers’ suggestions, and believe this change further improved our manuscript. Please find the enclosed letter addressing each of the reviewer's comments and the two versions (tracked-changes and clean) of the revised manuscript. The page and line references for the changes provided in this response letter are based on the tracked-changes version (marked copy) of the manuscript. We hope that you and the reviewers now find the revised paper suitable for publication.

Sincerely yours,

Jongmok Ha, MD

Department of Neurology, Samsung Medical Center, Sungkyunkwan University School of Medicine 81 Irwon-ro Gangnam-gu, Seoul, Republic of Korea (06351)

Tel: 82-2-3410-1426, Fax: 82-2-3410-3421

E-mail: jongmok3245@gmail.com

AND

Darda Chung, MD

Department of Neurology, Korea University Anam Hospital 

64, Goryeodae-ro 17ga-gil, Seongbuk-gu, Seoul, Republic of Korea

Tel: +82-10-3088-2370, Fax: +82-2-920-5893

E-mail: vogmag0107@gmail.com

Reviewer #1: Thank you for the article.

The results of this study are not much different than what we expected from literature and clinical experience.

1. There are of course confounding factors that plague all studies related to COVID studies including this one:

-The vaccines became available very quickly from a historical perspective. This makes interpretation of temporal associations between the pandemic and vaccination difficult. This is especially important if there is delay in reporting.

Response: Thank you for this relevant remark. Indeed, the vaccine rollout was very quick and concerns on the ‘appropriate room’ to calculate for temporal association may seem tight. During vaccine rollout, COVID-19 Vaccination Task Force Adverse Event Investigation Team was set up inside Korea Disease Control and Prevention Agency to lead a government-led surveillance program to record a comprehensive list of adverse events following immunization (AEFIs) (REF 1). Relevant medical personnel or the patient reported to the Korea Immunization Management System (KIMS), a web-based event monitoring system, with the earliest convenience (mostly within a week), and the task force, along with the local authorities, performed weekly meetings to go over the reported cases and determined whether the vaccination was safe to proceed. We believe this process shortened the delay in adverse event reporting significantly. Furthermore, our study design uses nationwide population data and does not rely on this reporting system in entirety. We are less subject to reporting bias as we used the total number of hemorrhagic stroke and CVST cases that were recorded during set time frame, not confined to the cases reported in the AEFI monitoring system.

REF 1. Hwang I, Park K, Kim TE, Kwon Y, Lee YK. COVID-19 vaccine safety monitoring in Republic of Korea from February 26, 2021 to October 31, 2021. Osong Public Health Res Perspect. 2021;12(6):396-402.

-The study relies on reported risk factors. This seems to be dependent on bias of health care professionals. For instance, oral contraceptives as a risk factor for CVST seem to be exaggerated. Not surprising then, that women are more often imaged for CVST than men, and less often that scans show CVST.

Response: We agree that the reported risk factors are hand-picked preferentially by the authors and may be biased in clinical perspective. To point this out, we have mentioned in the limitation on the list not being exhaustive (page 17, lines 258 – 260). Moreover, we tried to use these ‘conventional risk factors’ as correlation benchmarks to COVID-19 infection or COVID-19 vaccination to figure out if temporal course of the disease was more likely influenced by well-known risk factors or potential newcomers to the roster. It may be important to point out that these risk factors were by design not individually checked and recorded by health care professionals per incident case yet estimated as a prevalence within incident population using claims data (if the patient had at least once been prescribed an OC, had cancer, or had sepsis in the relevant time frame, he or she was added to the numerator) (page 7, lines 84 – 86). Furthermore, the selected risk factors are clinically relevant because they make up the big portion of total risk seen in clinical practice. Regarding oral contraceptives (OC), OCs are by far one of the most important risk factors in CVST, with odd’s ratio up to 7.5 in a systemic review (REF 1). Although the risk factor itself may have been subject to bias in terms of screening as the reviewer suggested, we doubt this would have been detrimental to our analysis as no sexual preponderance in incidence was observed (Table 1).

REF 1: Amoozegar F, Ronksley PE, Sauve R, Menon BK. Hormonal contraceptives and cerebral venous thrombosis risk: a systematic review and meta-analysis. Front Neurol. 2015 Feb 2;6:7. doi: 10.3389/fneur.2015.00007. PMID: 25699010; PMCID: PMC4313700.

-After becoming apparent that some vaccines are associated with ICH, they were withdrawn at least for younger populations in some European countries. How was that in Korea?

Response: Thank you for this comment. In Korea, the adenoviral vector-based ChAdOx1-S/nCoV-19 (Oxford-Astrazeneca) vaccine and Ad26. COV2.S (Janssen) vaccine was disapproved for ages 30 and under, after reports of thrombosis with thrombocytopenia (TTS), TTS-related CVST, and possibly ICH associated with TTS-related CVST. ICH collectively on the other hand (ICH and SAH combined, regardless of mechanism), despite reports of association with mRNA-based vaccines in some studies as referenced in our manuscript (REF 1, 2, 3), did not have an impact in steering policies on vaccine rollout. 

REF 1. Patone M, Handunnetthi L, Saatci D, et al. Neurological complications after first dose of COVID-19 vaccines and SARS-CoV-2 infection. Nat Med. 2021;27(12):2144-2153.

REF 2. Torabi F, Bedston S, Lowthian E, et al. Risk of thrombocytopenic, haemorrhagic and thromboembolic disorders following COVID-19 vaccination and positive test: a self-controlled case series analysis in Wales. Scientific reports. 2022;12(1):16406.

REF 3. Simpson CR, Shi T, Vasileiou E, et al. First-dose ChAdOx1 and BNT162b2 COVID-19 vaccines and thrombocytopenic, thromboembolic and hemorrhagic events in Scotland. Nature medicine. 2021;27(7):1290-1297.

-Definition of pandemic period can be discussed. By spring-summer of 2021 the number of severe cases had dropped drastically and for practical purposes the pandemic was mostly a common cold being diagnosed in asymptomatic individuals.

Response: As shown in figure 2, our country experienced a steep incline of new SARS-CoV-2 infection cases and deaths. The following are excerpts from Worldometer showing daily death counts due to COVID-19 (https://www.worldometers.info/coronavirus, REF 1), still showing relevant peaks throughout 2022. Excess mortality was also high during the same period (ourworldindata.org as reference, REF 2). The authors agree that since the introduction of omicron variant in December 2021, milder clinical manifestation, build-up of population immunity, and quick awareness of COVID symptoms had rendered the pandemic into common cold. However, it is unclear whether this also diminished the level of systemic immune response in patients. Regarding infections, many theories on mechanism have been suggested on how the virus could affect the patient. Aside from the fact that the virus could enter the bloodstream and directly influence the vasculature, secondary damage due to immune response (which is sometimes hyperinflammatory and detrimental) should not be neglected. As far as we are concerned, this immune response becomes increasingly robust with repeated exposures to the immune stimulus. Therefore, we believe that the pandemic period was reasonably set in our study, when confined to South Korea.

References)

REF 1. https://www.worldometers.info/coronavirus

REF 2. Edouard Mathieu, Hannah Ritchie, Lucas Rodés-Guirao, Cameron Appel, Charlie Giattino, Joe Hasell, Bobbie Macdonald, Saloni Dattani, Diana Beltekian, Esteban Ortiz-Ospina and Max Roser (2020) - "Coronavirus Pandemic (COVID-19)". Published online at OurWorldInData.org. Retrieved from: 'https://ourworldindata.org/coronavirus' [Online Resource]

2. The authors claim that the decrease in incidence of SAH might be related to less access to health care. While there is evidence for some conditions such as appendicitis in March-May 2020 in Europe, by June 2020 it was business as usual for us in Europe. In the case of appendicitis as an example there was increased incidence of complications in the ensuing months with 20-40% presenting with abscess. How would that be for SAH if there is a delay?

Response: We thank the reviewer for steering our view towards the aftermath of undetected acute disorders. SAH is in most cases an urgent disorder with immediate case fatality rate hovering around 40% if left untreated and reaching above 50% considering morbidity risks within 6 months (REF 1, 2). Our guess is that sudden deaths or death by unknown cause may have increased during the same period as a complication. This is an excellent segue to a follow-up study to confirm if our speculations were indeed valid. We included this feedback in our discussion section (page 15, lines 209 – 212).

REF 1. Lantigua H, Ortega-Gutierrez S, Schmidt JM, Lee K, Badjatia N, Agarwal S, Claassen J, Connolly ES, Mayer SA. Subarachnoid hemorrhage: who dies, and why? Crit Care. 2015 Aug 31;19(1):309. doi: 10.1186/s13054-015-1036-0. PMID: 26330064; PMCID: PMC4556224.

REF 2. Mahlamäki K, Rautalin I, Korja M. Case Fatality Rates of Subarachnoid Hemorrhage Are Decreasing with Substantial between-Country Variation: A Systematic Review of Population-Based Studies between 1980 and 2020. Neuroepidemiology. 2022;56(6):402-412. doi: 10.1159/000526983. Epub 2022 Oct 25. PMID: 36282049.

3. CVST caused by vaccine is much more serious than a CVST caused by natural infections from an ethical point of view. This is also important since the risk for thrombosis with COVID infection seems to have been exaggerated based on what we know.

Response: We agree that in ethical standpoint, CVST caused by vaccine can be a serious problem. There is evidence of both an increased CVST incidence due to SARS-CoV-2 infection (REF 1) and SARS-CoV-2 vaccination in the literature (REF 2). However, there is no head-to-head comparison on both etiologies in the literature. Mechanism-wise, SARS-CoV-2 infection is more prone to causing ischemic stroke due to large vessel occlusion due to vasculopathies than hypercoagulability (REF 3). Moreover, excess increase in thrombotic complications including CVST has been reported by the viral vector-based vaccine company and a withdrawal process commenced after diminishing of demands as a response to this allegation (REF 4).

REF 1. Ohaeri C, Thomas DR, Salmon J, Cottrell S, Lyons J, Akbari A, Lyons RA, Torabi F, Davies GG, Williams C. Comparative risk of cerebral venous sinus thrombosis (CVST) following COVID-19 vaccination or infection: A national cohort study using linked electronic health records. Hum Vaccin Immunother. 2022 Nov 30;18(6):2127572. doi: 10.1080/21645515.2022.2127572. Epub 2022 Oct 27. PMID: 36302124; PMCID: PMC9746546.

REF 2. Krzywicka K, Heldner MR, Sánchez van Kammen M, van Haaps T, Hiltunen S, Silvis SM, Levi M, Kremer Hovinga JA, Jood K, Lindgren E, Tatlisumak T, Putaala J, Aguiar de Sousa D, Middeldorp S, Arnold M, Coutinho JM, Ferro JM. Post-SARS-CoV-2-vaccination cerebral venous sinus thrombosis: an analysis of cases notified to the European Medicines Agency. Eur J Neurol. 2021 Nov;28(11):3656-3662. doi: 10.1111/ene.15029. Epub 2021 Aug 4. PMID: 34293217; PMCID: PMC8444640.

REF 3. Wijeratne T, Sales C, Karimi L, Crewther SG. Acute Ischemic Stroke in COVID-19: A Case-Based Systematic Review. Front Neurol. 2020 Sep 25;11:1031. doi: 10.3389/fneur.2020.01031. PMID: 33101164; PMCID: PMC7546832.

REF 4. https://www.ema.europa.eu/en/human-regulatory-overview/public-health-threats/coronavirus-disease-covid-19/covid-19-public-health-emergency-international-concern-2020-23/withdrawn-applications-products

Reviewer #2: The authors present a retrospective cohort study evaluating the rates of SAH, ICH, and CVST in South Korea from 2007 - 2022. Rates of cancer, sepsis, hormone therapy, anticoagulation, aneurysm, HTN, renal disease, liver disease, etc. were also reported. The manuscript concludes that SAH and ICH rates declined during the pandemic, while CVST increased. The authors posit the conjecture that this may be related SARS-COV vaccination.

1. Please, replace the ICD-10 codes in your figures / tables with the names of the diseases.

Response: Thank you for this comment. We replaced the ICD-10 codes in our figures and tables with the name of the disease as suggested by the reviewer.

2. I would recommend expanding your discussion around CVST. Particularly, it would be helpful to further discuss the limitations of your data (small N for CVST events versus model variable count, risk factors for CVST missing from dataset, lack of post vaccination push data...) and what additional studies you feel would be needed to better address this question.

Response: Thank you for this constructive feedback. We have expanded the list of limitations as recommended by the reviewer and discussed future studies that may help address these limitations (page 17, lines 260 – 265).

---

## [Editor Report · Decision Letter 1]

18 Jun 2024

Deciphering changes in the incidence of hemorrhagic stroke and cerebral venous sinus thrombosis during the coronavirus disease 2019 pandemic: a nationwide time-series correlation study

PONE-D-24-10134R1

Dear Dr. Ha,

We’re pleased to inform you that your manuscript has been judged scientifically suitable for publication and will be formally accepted for publication once it meets all outstanding technical requirements.

Kind regards,

Sonu Bhaskar, MD PhD

Academic Editor

PLOS ONE

Additional Editor Comments (optional):

Thank you for submitting the revised version of your manuscript. No further comments. I am pleased to accept the manuscript in its current form.
---

## [Editor Report · Acceptance letter]

8 Aug 2024

PONE-D-24-10134R1 

PLOS ONE

Dear Dr. Ha, 

I'm pleased to inform you that your manuscript has been deemed suitable for publication in PLOS ONE. Congratulations! Your manuscript is now being handed over to our production team.

Kind regards, 

on behalf of

Dr. Sonu Bhaskar 

Academic Editor

PLOS ONE